

# Antioxidant activity, α-glucosidase inhibition and phytochemical profiling of *Hyophorbe lagenicaulis* leaf extracts

James William[1], Peter John[1], Muhammad Waseem Mumtaz[2], Ayoub Rashid Ch[1], Ahmad Adnan[1], Hamid Mukhtar[3], Shahzad Sharif[1], Syed Ali Raza[1] and Muhammad Tayyab Akhtar[3]

[1] Department of Chemistry, Government College University Lahore, Lahore, Pakistan
[2] Department of Chemistry, University of Gujrat, Gujrat, Pakistan
[3] Institute of Industrial Biotechnology, Government College University Lahore, Lahore, Pakistan

## ABSTRACT

**Background:** Diabetes mellitus type II (DMT-2) is a widely spread metabolic disorder both in developed and developing countries. The role of oxidative stress is well established in DMT-2 pathogenesis. The synthetic drugs for DMT-2 are associated with serious side complications. Antioxidant and α-glucosidase inhibitory actions of phytochemicals from various plant species are considered as an alternative to synthetic drugs for DMT-2 management. The present study aimed to evaluate the antioxidant activity, α-glucosidase inhibitory potential and phytochemical profiling of *Hyophorbe lagenicaulis*.

**Methods:** The total phenolic and flavonoid contents, in vitro antioxidant activity (α, α-diphenyl-β-picrylhydrazyl (DPPH) free radical scavenging and phosphomolybdenum method) and α-glucosidase inhibition of ultrasonicated hydroethanolic *H. lagenicaulis* leaf extracts were determined spectrophotometrically. The results of DPPH assay and α-glucosidase inhibition were reported in terms of $IC_{50}$ value. The phytochemical profiling was accomplished by UHPLC-Q-TOF/MS/MS technique.

**Results and Discussion:** Findings leaped 60% ethanolic extract as rich fraction regarding total phenolic and flavonoid contents. The 60% ethanolic fraction was a promising source of natural antioxidants and α-glucosidase inhibitory agents as indicated by anti-radical and enzyme inibitory activities. Kaempferol, rutin, hesperetin 5-O-glucoside, kaempferol-coumaroyl-glucoside, luteolin 3-glucoside, Isorhamnetin-3-O-rutinoside, trimethoxyflavone derivatives and citric acid were identified by UHPLC-Q-TOF-MS/MS. These compounds were believed to be responsible for the strong antioxidant and enzyme inhibitory activity of plant extracts. The extensive metabolite profiling of *H. lagenicaulis* was carried out the first time as never reported previously. The *H. lagenicaulis* might be an appropriate choice to manage diabetes mellitus in an alternate way. The findings may be further exploited extensively for toxicity evaluation to proceed with functional food development having antidiabetic attributes.

Corresponding authors
Peter John, peterjohn@gcu.edu.pk
Muhammad Waseem Mumtaz, muhammad.waseem@uog.edu.pk

## INTRODUCTION

Diabetes is one of the leading non-infectious diseases with multiple side complications characterized by persistent hyperglycemia due to reduced insulin secretion or action (*Schwartz et al., 2016*). More than 90% diabetic patients are suffering from diabetes mellitus type 2 (DMT-2) and it is estimated that by 2,035 the expansion of DMT-2 will result in 592 million diabetic patients worldwide (*Guariguata et al., 2014*). The huge expansion rate of DMT-2 is now considered a socio-economic burden and covers about 10% of the total health care expenditures in many countries (*Pari & Saravanan, 2007*). The modern lifestyle and dietary habits are among the key factors responsible for the DMT-2 progression. These factors are associated with the production of reactive oxygen species (ROS) which in excess may generate the state of oxidative stress. The role of ROS and oxidative stress in DMT-2 progression is evident from scientific studies (*Charokopou et al., 2016*; *Sami et al., 2017*). The mechanism of action through which oxidative stress contributes to DM pathogenesis is not fully clear. However, the hyperglycemia is reported to produce free radicals which impair the function of antioxidant enzymes in plasma (*Asmat, Abad & Ismail, 2016*). The oxidative stress also impairs the insulin secretion, alteration in glucose uptake, abnormal glucose release form liver and mediation of metabolic pathways (*Akash et al., 2011*). The natural antioxidants in the body are mainly enzymes which include superoxide dismutase, glutathione peroxidases and reductases. The vitamin A and E are also important endogenous antioxidants which substantially reduce the level of oxidative stress (*Madhikarmi & Murthy, 2014*). The glycation of antioxidant enzymes also alters the structure based function of enzymes to increase the chances of damage by ROS (*Singh et al., 2014*). The lipid oxidation and glutathione metabolism impairments are used as biomarkers for DM and oxidative stress usually alters this to initiate DMT-2. Antioxidants are believed to encounter ROS to reduce level of oxidative stress to prevent development of DMT-2. High levels of ROS are involved in glycation of proteins, lipid peroxidation and glucose oxidation and these collectively impart in DMT-2 development and related disorders (*Asmat, Abad & Ismail, 2016*). The elimination of ROS or reduction in the level of oxidative stress may diminish the chances of DMT-2 pathogenesis and prolongation by improving the intra cellular antioxidant defense (*Ceriello & Testa, 2009*). The antioxidants based therapy is considered as promising approach to treat DMT-2 as antioxidants effectively scavenge the free radicals and ROS to prevent DM pathogenesis and related complications (*Rahimi-Madiseh et al., 2016*).

Many synthetic drugs are available to manage diabetes but the side effects associated with these compounds are a matter of keen concern (*Chaudhury et al., 2017*). Health related drawbacks of synthetic medicines emphasize on the need to develop alternate treatments for DMT-2. Plants are a rich source of natural, safe and potent phytochemicals of medicinal attributes to treat chronic ailments. The many natural bioactive ingredients in plants are also associated with α-glucosidase inhibition. The α-glucosidase inhibitors restrict the hydrolysis of carbohydrates (Oligosaccharides, trisaccharides and disaccharides) in the intestine and slow down their absorption, hence limit the postprandial glucose level (*Rouzbehan et al., 2017*). Plants also contain natural antioxidants which encounter the

exceeding ROS levels in body to reduce the risks of DMT-2 pathogenesis and progression (*Zaid et al., 2015*). Therefore search for α-glucosidase inhibitors from plants for DMT-2 management is workable, safe, reliable and cost effective approach.

Family *Arecaceae* contains more than 189 genera, 3,000 species and some species are very rich in antioxidants and other functional molecules. Plants from family *Arecaceae* have been studied for their potential medicinal use however many species are still needed to explore for their hidden biological and pharmacological role (*Govaerts & Dransfield, 2005*). *Hyophorbe lagenicaulis* of family *Arecaceae* is among such plants which are not completely studied for their potential biological activities (*Elgindi et al., 2016*). Inspite of effective role of *H. lagenicaulis* in folk medicines, no scientific evidence is available on biological activities and phytochemical distribution in this plant. The current work was performed to evaluate the in vitro antioxidant and antidiabetic potential of *H. lagenicaulis*. The metabolite profiling of leaf extract of *H. lagenicaulis* was also carried out.

## MATERIAL AND METHODS

### Collection of plant material
The plant material was collected from Lahore, Pakistan and was identified from the Department of Botany, GC University Lahore.

### Green extract preparation
Quenching of fresh leaves was done in liquid nitrogen and grinded to a fine powder to enhance the surface area. The obtained powder was lyophilized using a freeze-dryer (Christ Alpha 1-4 LD; Osterode am Harz, Germany) and subjected to hydroethanolic solvent compositions (Ethanol, 100%, 80%, 60%, 40%, 20%) for 48 h. Mixtures were sonicated at soniprep 150 disintegrator below 10 °C. Samples were shaken for 2 h and filtered. The excess solvent from the filtrate was removed under the vacuum on rotary evaporator at 40 °C. The extracts were again freeze dried for 48 h. Extract yields (%) were calculated and extracts were stored at −80 °C till further use.

### Determination of total phenolic and flavonoid contents
Total phenolic contents (TPC) of freeze dried leaf extracts were determined by Folin Ciocalteu reagent method with slight modification in previously reported scheme (*Zhishen, Mengcheng & Jianming, 1999*). The plant extract (one mg) was dissolved in methanol (one mL) and 0.25 µL of this was added to one mL of Folin Ciocalteu reagent. Then two mL of 10% solution of $Na_2CO_3$ followed by addition of two mL distilled water. The resultant mixture was stayed for 120 min at ambient conditions of temperature. The absorbance was noted at 765 nm. Standard curve of gallic acid was also drawn. Results were expressed as gallic acid equivalent (GAE) mg/g dried extract (*Zengin et al., 2010*).

Total flavonoid contents (TFC) were determined by $AlCl_3$ colorimetric method. The 0.1 mg of plant extract was dissolved in methanol (two mL) and added by five mL of distilled water. Then 0.5 mL of $NaNO_2$ (5%) was added to the mixture followed by the addition of 10% $AlCl_3$ solution. After 10 min NaOH (1 molar) was added to resultant

mixture and after vigorous shaking, the absorbance was measured at 510 nm. The results were expressed as rutin equivalent mg/g dried extract (RE mg/DE) (*Zhishen, Mengcheng & Jianming, 1999*).

## Antioxidant activities

Antioxidant potential of extracts was determined by DPPH scavenging assay as reported previously with little modification (*Fki, Allouche & Sayadi, 2005*). The one mg plant extract was dissolved in methanol. The 50–200 ppm of these dilutions were added to 10 mL of DPPH solution (0.001 molar). After 30 min incubation in the dark at room temperature, absorbance was measured at 520 nm. Butylated hyroxyanisole (BHA) was used as standard for comparison. All the measurements were carried out in triplicate and standard deviation was applied.

Phosphomolybdenum complex formation method was used as per previously reported method with minute modifications (*Prieto, Pineda & Aguilar, 1999*). Initially, 250 µg/mL of each extract was mixed with a solution composed of sulphuric acid (0.6M), ammonium molybdate (four mM) and 28 mM sodium phosphate. The mixtures and blank solution both were incubated at 95 °C for 90 min at water bath. After cooling, absorbance was noted at 695 nm wavelength. Ascorbic acid standard curve was drawn and butylated hyroxyanisole was used as standard antioxidant. The results were represented as ascorbic acid equivalent per gram dried plant extract (AE/g DE).

## Anti-α-glucosidase activity

Inhibition potential of extracts against α-glucosidase was measured to evaluate in vitro antidiabetic potential. Various concentrations of extarcts (200 ppm) were added to phosphate buffer (70 µL of 50 mM) followed by addition of α-glucosidase (one unit/mL). After 10 min incubation at 37 °C, five mM of ρ-nitrophenolglucopyranoside was added and absorbance was noted at 405 nm after 30 min. Acarbose was used as standard reference and results were represented as $IC_{50}$ (µg/mL) values for each extract (*Jabeen et al., 2013*).

$$\% \, \text{Inhibition} = \frac{Ab - As}{Ab} \times 100$$

Where, Ab is absorbance of blank, As is absorbance of Sample.

## The α-amylase inhibition activity

The 250 µL of each extract (1.0–10 mg/mL) were added to 0.02M sodium phosphate buffer containing porcine α-amylase (0.5 mg/mL). The reaction mixture was incubated for 10 min at 25 °C. The dinitrosalicylic acid was added to the mixture to stop the reaction. The reaction mixtures were further incubated for a time period of 5 min and diluted with distilled water to note absorbance at 540 nm. A control (no extract) was also run and acarbose was used as standard enzyme inhibitory substance (*Kazeem, Adamson & Ogunwande, 2013*). The % inhibition was calculated by following relationship;

**Table 1 Extract yields, TPC and TFC from leaves of *H. lagenicaulis* fractions.**

| Solvent composition | Extract yield (%) | TPC in mg GAE/g DE | TFC in mg Rutin/g DE |
|---|---|---|---|
| 20% ethanol | 14.31 ± 0.2[de] | 78.09 ± 1.36[d] | 68.94 ± 1.61[e] |
| 40% ethanol | 16.33 ± 0.32[c] | 109.63 ± 1.67[c] | 92.02 ± 1.72[d] |
| 60% ethanol | 20.46 ± 0.25[a] | 178.56 ± 1.47[a] | 133.96 ± 1.19[a] |
| 80% ethanol | 18.05 ± 0.13[b] | 144.67 ± 2.31[b] | 115.51± 0.90[b] |
| 100% ethanol | 15.10 ± 0.15[d] | 109.62 ± 0.44[c] | 100.90 ± 1.59[c] |

**Note:**
Results were represented with standard deviation values (±) and significant level was indicated by letter as superscript.

$$\% \text{ Inhibition} = \frac{Ab - As}{Ab} \times 100$$

Where, Ab is absorbance of blank, As is absorbance of Sample.

Results were represented as $IC_{50}$ (µg/mL) values for each extract.

## UHPLC-Q-TOF-MS/MS analysis

Plant extract was extracted with suitable solvent and filtered with plastic filter having 0.45 µm pore size. The filtered extract sample was subjected to UHPLC-Q-TOF-MS/MS (AB Sciex 5600-1, equipped with Eksigent UHPLC system). The scanning range of 50–1,200 $m/z$ for MS/MS (negative ionization mode), column Thermo Hypersil Gold (100 mm × 2.1 mm × 3 µm), gradient mobile phase composition (water and acetonitrile) each having 0.1% formic acid and five mM ammonium formate was used. Gradient programming started from 10% acetonitrile to 90% acetonitrile with mobile phase flow rate of 0.25 mL/min. Desolvation temperature (TEM) was 500 °C and ion spray voltage was −4,500 V.

## Statistical analysis

The experimental findings were evaluated for statistical significance by using Statistix 10.0 software. Analysis of variance was used to compare variations in treatment means to assess efficacy of treatments.

# RESULTS

## Extract yields (%), TPC and TFC

The extract yields, TPC and TFC with respect to solvent system used for extraction are given in Table 1. It was revealed by the findings that solvent influenced extract yields significantly. Maximum extract yield 20.46% ± 0.25[a]% was given by 60% ethanol and second highest yield 18.05% ± 0.13[b]% was obtained with 80% ethanol. Solvent composition not only affected the extract yields but also TPC and TFC. Maximum TPC of 178.56 ± 1.47[a] mg GAE/g DE were recovered with 60% ethanol. Similarly highest TFC of 133.96 ± 1.19[a] mg Rutin/g DE were extracted with 60% ethanol. Efficiency of 60% ethanolic extract was significantly higher than other solvent fraction for extract yield, TPC and TFC respectively ($\rho < 0.05$).

## Antioxidant activities

The DPPH radical scavenging in terms of IC-50 value (µg/mL) by plant extracts in comparison with BHA is represented as Fig. 1. Maximum radical scavenging among

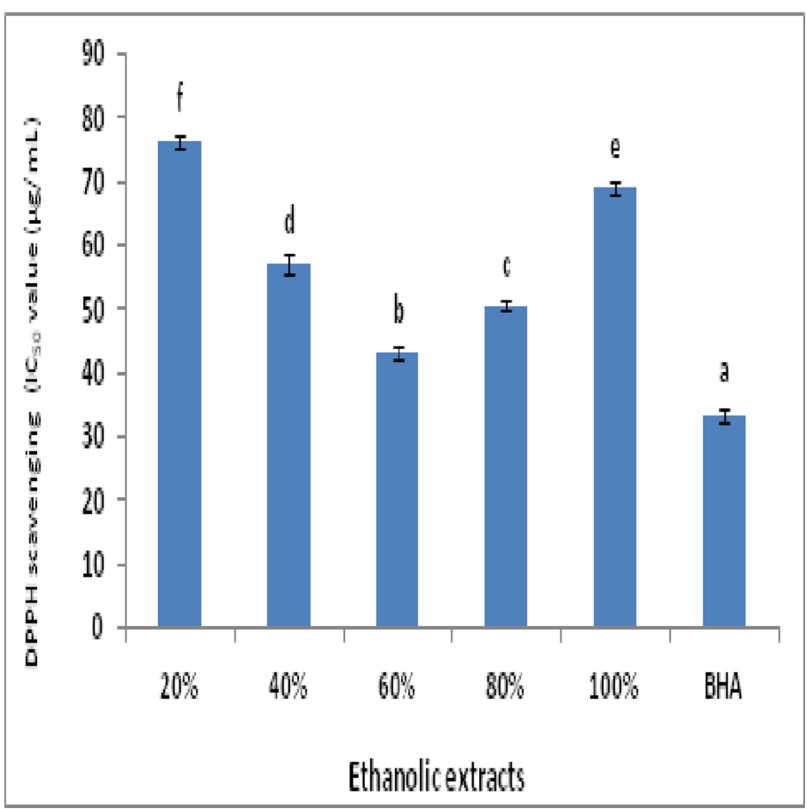

**Figure 1 DPPH radicals scavenging activity in terms of IC-50 value for plant extracts and BHA (a, reference; b–f, ascending order of IC-50).**

extracts was exhibited by 60% ethanolic extract with minimum IC-50 value of 43.11 ± 0.96 µg/mL.

Total antioxidant power of extracts was determined by phosphomolybdenum methods. This method involved the reduction of Mo (VI) to Mo (V) with characteristic color change due to complex formation. This assay is widely used to evaluate the total antioxidant power of plant extracts and compounds (*Prieto, Pineda & Aguilar, 1999*; *Rani et al., 2018*).

The results of assay were represented in Fig. 2. Findings unveiled that 60% ethanolic fraction exhibited maximum antioxidant power with value of 239.33 ± 3.78[b] (ASE/g PE) followed by 80% ethanolic extract (189.33 ± 2.51[c] ASE/g DE). Statistical analysis indicated that antioxidant power of BHA was significantly higher than all extracts ($\rho < 0.05$). However 60% ethanolic extract was most potent among all plant extracts ($\rho < 0.05$) regarding antioxidant potential.

## Anti-α- glucosidase activity

Inhibition of α-glucosidase enzyme reflects the in vitro antidiabetic potential which is determined spectrophotometrically. The IC-50 values (µg/mL) of plant extracts and standard drug acarbose are represented in Fig. 3. The comparison of extracts showed that 60% ethanolic extract among all fractions possessed maximum α-glucosidase inhibition with IC-50 value of 41.25 ± 1.25 µg/mL. The standard drug acarbose exhibited lowest IC-50 value of 25.50 ± 0.45 µg/mL which was significantly lower than all extracts

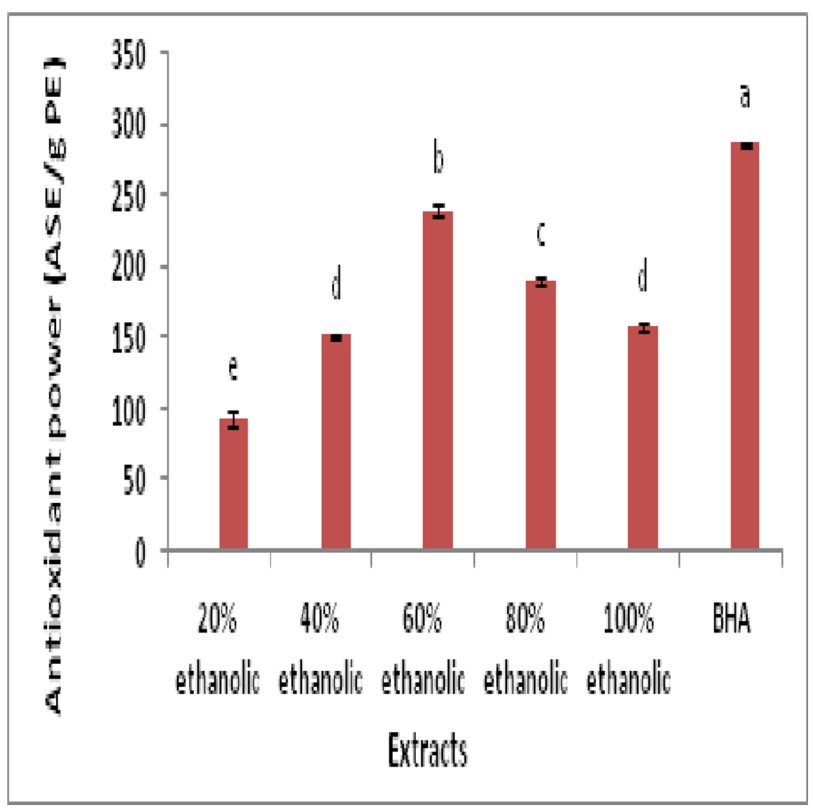

**Figure 2 Antioxidant power assay (ASE/g PE) for determination of antioxidant activity of plant extracts and BHA (a, reference; b–e, descending order).**

($\rho < 0.05$). Statistical comparison further indicated that 60% ethanolic extract was significantly better than remaining extracts ($\rho < 0.05$).

### The α- amylase inhibition

The results of α-amylase inhibition are given in Fig. 4. The 60% ethanolic extract (IC-50 60.58 ± 3.24 μg/mL) exhibited highest α-amylase inhibition followed by 80% ethanolic extract with IC-50 value of 77.57 ± 2.25 μg/mL. The 20% ethanolic extract exhibited least inhibition of enzyme activity as indicated by IC-50 value (114.00 ± 1.88 μg/mL). The standard compound acarbose showed highest α-glucosidase inhibition with IC-50 value of 43.37 ± 0.75 μg/mL. Statistical analysis revealed that 60% ethanolic extract was the most potent α-glucosidase inhibitory fraction.

### UHPLC-Q-TOF-MS/MS analysis

The efficacy of 60% ethanolic extract regarding antioxidant and α-glucosidase inhibition emphasized to explore this fraction for metabolite identification. So the 60% ethanolic extract was subjected to UHPLC-Q-TOF-MS/MS analysis and identified compounds along with their retention time (Rt), fragment ions and molecular formula are listed in Table 2. The main chromatogram of UHPLC separation is shown as Fig. 5. The fragmentation pattern of identified compounds is represented in Fig. 6. Citric acid appeared at Rt 1.603 min with characteristic parent ion peak at $m/z$ 191[M-H]$^-$ and daughter ion peak at

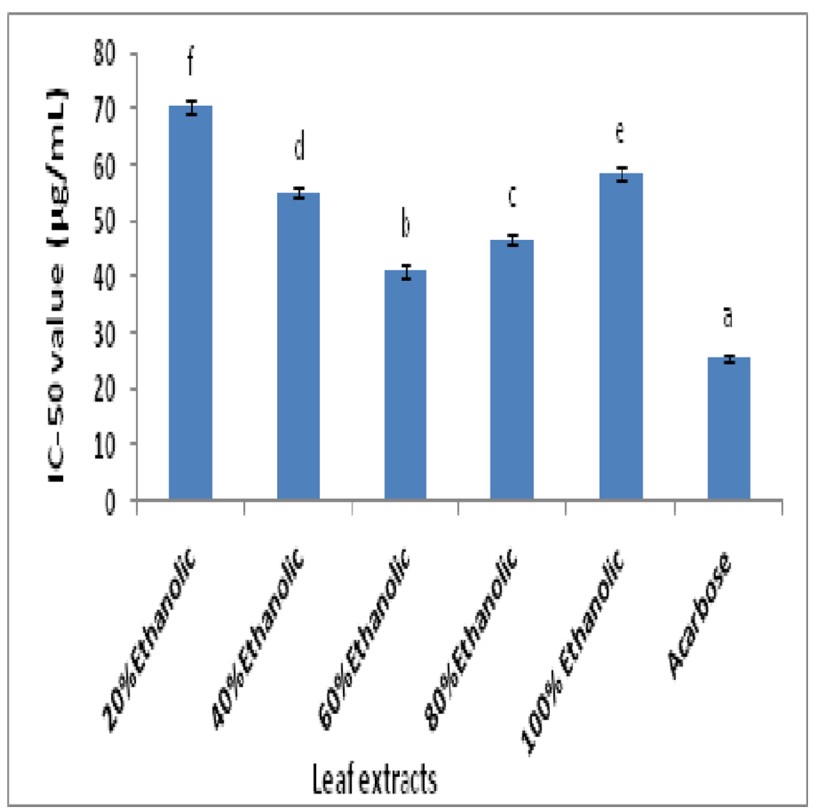

**Figure 3 The IC-50 values for α-glucosidase inhibitory potential of extract fractions and acarbose (a, reference; b–f, ascending order of IC-50).**

$m/z$ 111$[M-CO_2-H_2O]^-$. Trimethoxy flavone derivative arrived at Rt 8.972 min with peaks at $m/z$ 635, $m/z$ 609 and $m/z$ 300. Kaempferol was recorded at Rt 9.110 min with $m/z$ 285. The fragment ion peaks at $m/z$ 151 and $m/z$ 93 due to removal of fragment of 134 amu and phenoyl moiety respectively. Rutin was recorded at Rt 9.27 min with parent ion peak $m/z$ 609. The fragment ion having $m/z$ 300 appeared due to removal of moiety of mass 309 amu. The fragment ion of $m/z$ 271 was produced due to removal of 29 amu from fragment ion $m/z$ 300. Peak at Rt 9.689 was identified as kaempferol-coumaroyl-glucoside with parent peak at $m/z$ 593. The removal of coumaroyl glucoside produced fragment ion peak at $m/z$ 285. Leuteolin 3-glucoside was identified at Rt 9.724 with parent ion peak at $m/z$ 447. Removal of glucose from parent ion generated leuteolin characteristic peak at $m/z$ 285. Hesperetin 5-O-glucoside was identified at Rt 9.433 with parent peak at $m/z$ 463$[M-H]^-$ and fragment ions at $m/z$ 301$[M-glucose-H]^-$, $m/z$ 271$[m/z$ 301-$CH_2$O-H$]^-$ $m/z$ 255$[m/z$ 301-$C_2H_2$O-H$]^-$, $m/z$ 149$[m/z$ 255-$C_6H_2O_2$-H$]^-$ respectively. Isorhamnetin 3-O-rutinoside appeared at Rt 9.995 with $m/z$ 623. The further collision resulted in fragment ions $m/z$ 315$[M-318-H$ amu$]^-$, $m/z$ 300 $[m/z$ 315-$CH_3$-H$]^-$ and $m/z$ 284$[m/z$ 315-$CH_3$O-H$]^-$.

## DISCUSSION

The polarity of the solvent system used for extraction might be the decisive factor for enhanced productivity (*Chew et al., 2011*). Phenolic and flavonoid compounds present in

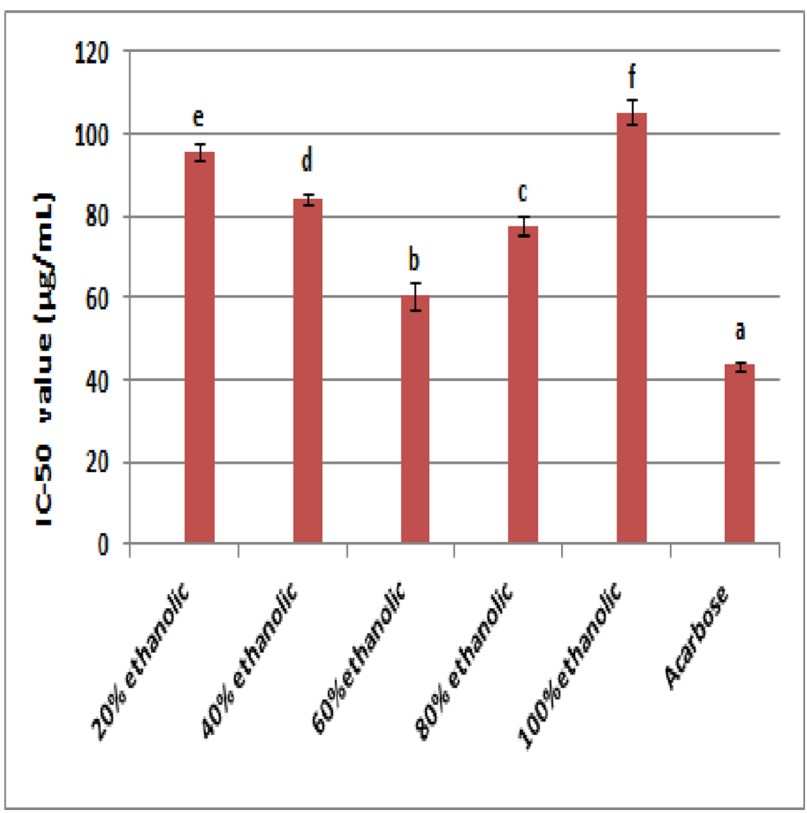

**Figure 4 The IC-50 values for α-amylase inhibitory potential of extract fractions and acarbose (a, reference; b–f, ascending order of IC-50).**

**Table 2 Peak assignments for identified compounds by UHPLC-MS/MS in negative mode.**

| Sr. No | Name of compound | Rt (min) | Molecular ion peak ($m/z$) | Main fragments ion ($m/z$) | Molecular formula |
|---|---|---|---|---|---|
| 1 | Citric acid | 1.603 | 191 | 111 | $C_6H_8O_7$ |
| 2 | Trimethoxy flavone derivative | 8.972 | 773 | 635, 609, 300 | $C_{40}H_{38}O_{16}$ |
| 3 | Kaempferol | 9.110 | 285 | 151, 93 | $C_{15}H_{10}O_6$ |
| 4 | Rutin | 9.27 | 609 | 300, 271 | $C_{27}H_{30}O_{16}$ |
| 5 | Hesperetin 5-O-glucoside | 9.433 | 463 | 301, 300, 271, 97 | $C_{22}H_{24}O_{11}$ |
| 6 | Kaempferol-coumaroyl-glucoside | 9.689 | 593 | 285, 284, 255 | $C_{31}H_{30}O_{12}$ |
| 7 | Luteolin 3-glucoside | 9.724 | 447 | 285, 284, 255, 227 | $C_{21}H_{20}O_{11}$ |
| 8 | Isorhamnetin-3-O-rutinoside | 9.995 | 623 | 543, 527, 427, 315, 314 | $C_{21}H_{36}O_{21}$ |

plants are associated with medicinal properties. Higher concentration of both phenolics and flavonoids triggers the pharmaceutical and biological attributes of a particular plant (*Baba & Malik, 2015*; *Shreshtha et al., 2017*). The discrimination in TPC and TFC was probably due to solvent polarity interaction with heterogeneous structural features of phytochemicals. The promising antioxidant activity (DPPH and phosphomolybdenum complex method) was exerted by extracts. The proton transfer from phenolic compound
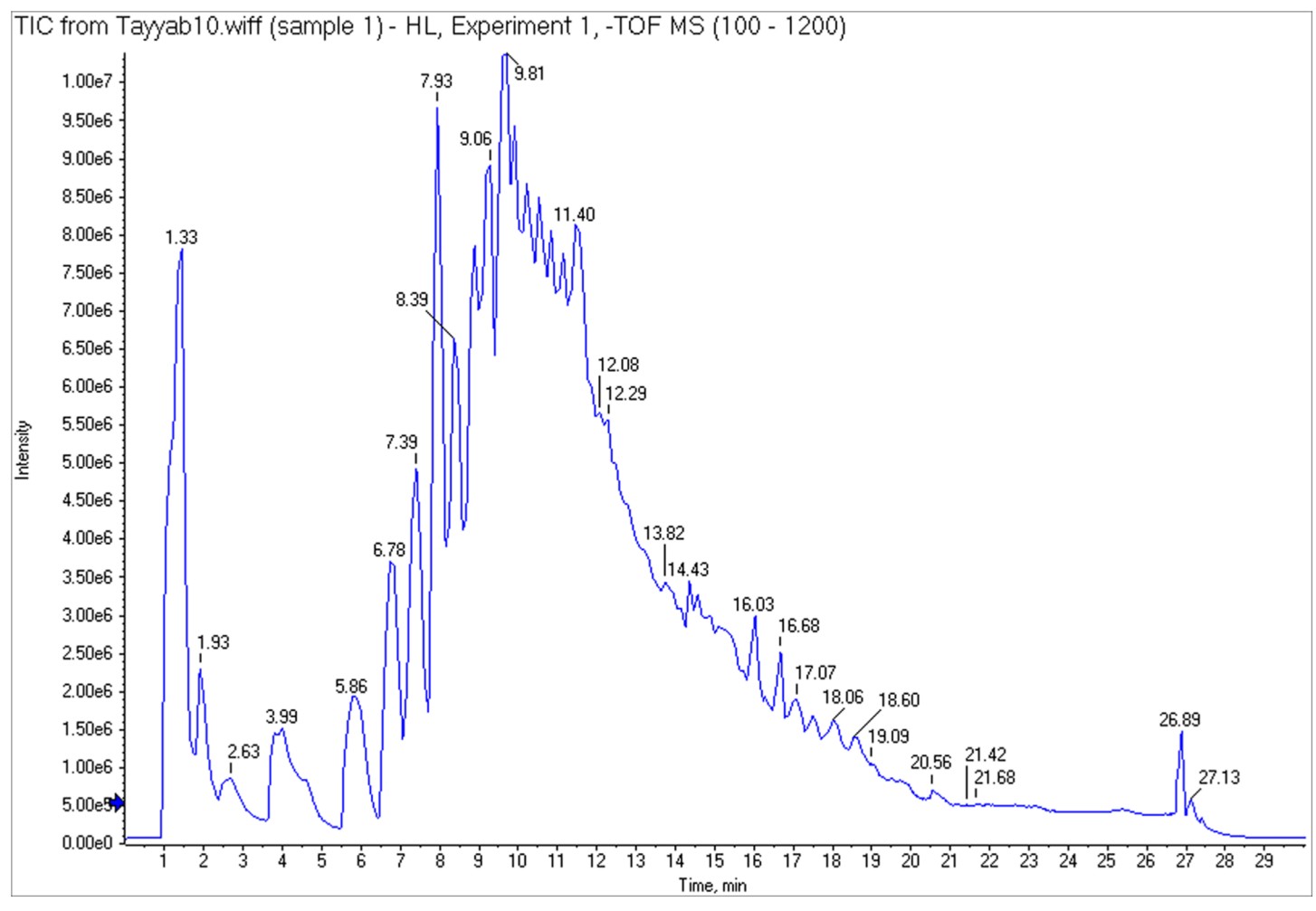

**Figure 5 Main chromatogram of *H. lagenicaulis* (UHPLC) indicating the peaks of eluted compounds.**

to DPPH free radical was reported as the possible mechanism in free radical scavenging process (*Liang & Kitts, 2014*). The proton and electron transfer from antioxidants to Mo(VI) resulted in loss of blue coloration to decrease the intensity of absorbance. The reduction in absorbance was used to assess the antioxidant potential (*Prieto, Pineda & Aguilar, 1999*). The antioxidant potential of plant extracts is directly proportional to the concentration and quality of phytochemicals including phenolics and flavonoids (*Liew et al., 2018*). The high α-glucosidase inhibitory action of 60% ethanolic extract was probably due to its high phenolic and flavonoid contents. Many plant based bioactive ingredients are well known α-glucosidase inhibitors and this potential may be exploited to manage DMT-2 (*Yin et al., 2014*). The 60% ethanolic extract might be a rich source of natural α-glucosidase inhibitors. The anti α-glucosidase activity exhibited by natural inhibitors was probably due to active site occupation by a particular inhibitor molecule to restrict the mode of enzymatic action by structural modification (*Martinez-Gonzalez et al., 2017*). Citric acid and other identified compounds of flavonoid origin are well known

**Figure 6 Fragmentation pattern of identified compounds with respective *m/z* values.**

antioxidants (*Rostamzad et al., 2011*). The antioxidant activities of these identified compounds were due to the interaction of phenolic groups and related structural features with ROS and other free radicals (*Aadesariya, Ram & Dave, 2017*). The antioxidant activities of plants are known as a decisive factor to control ROS and to eliminate state of oxidative stress. The reduction in oxidative stress can improve the physiological status to avoid DM pathogenesis and prolongation (*Singh, Parasuraman & Kathiresan, 2018*). The identified compounds in addition to antioxidant potential, were also reported to be associated with antidiabetic potential. The kaempferol and rutin, both flavonoids were proved to inhibit the α-glucosidase activity to control glucose homeostasis (*Pereira et al., 2011*). Another study reported the comparative evaluation of anti- α-glucosidase activity of kaempferol and quercetin. The findings revealed that kaempferol due to low IC-50 value was more efficient α-glucosidase inhibitor than quercetin (*Dewi & Maryani, 2015*). Rutin is a widely distributed polyphenolic flavonoid of plants. Previous reports also

highlighted the effective contribution of rutin against α-glucosidase activity, diabetes and obesity (*Jo et al., 2009*; *Habtemariam & Lentini, 2015*). Luteolin and its derivatives were reported to have promising anti- α-glucosidase inhibition potential even higher than acarbose suggesting it as a functional tool to control postprandial hyperglycemia (*Kim, Kwon & Son, 2000*). Isorhamnetin 3-O-rutinoside a flavonoid, was reported as a perfect α-glucosidase inhibitor with significantly low IC-50 values (*Yin et al., 2014*). The α-glucosidase inhibitory activity of acarbose is well established but some gastrointestinal problems also lie with it (*Van De Laar, 2008*). The acarbose was proved as a competitive inhibitor of α-glucosidase while plant extracts having phenolics were reported to possess non-competitive inhibition of dietary enzyme. The non-competitive mode provided multiple site interactions of phenolics with α-glucosidase rather than limited binding as in case of acarbose. In contrast to acarbose, α-glucosidase inhibition by phenolics in plant extracts does not depend upon the substrate concentration (*Zhang et al., 2015*). A recent study evaluated the phenolic contents, flavonoids, antioxidant and antidiabetic activities of hydroethanolic leaf extract of *Conocarpus erectus*. The study supported the linkage between polyphenol based antioxidant activity and hypoglycemic potential of extract (*Raza et al., 2018*). Another investigation revealed that phytochemicals from plants not only reduce the blood glucose level during diabetes but also improves the hematological parameters (*Sudasinghe & Peiris, 2018*). The antioxidant and anti-α-glucosidase potentials of *H. lagenicaulis* extracts were probably due to synergic behavior commonly observed with biologically functional plant extracts (*Adamska-Patruno et al., 2018*). The presence of high value bioactives in plants supports the efforts being made in search of safe and healthy therapeutic approaches for DMT-2 management. The study confirmed the antioxidant and antidiabetic potential of *H. lagenicaulis* leaves. The findings may be helpful to move for the reduction in socioeconomic burden, build by DMT-2.

## CONCLUSION

The promising antioxidant activity and α-glucosidase inhibition by *H. lagenicaulis* plant extracts were probably due to the presence of kaempferol, rutin, isorhamnetin and luteolin derivative. The findings provided us with leads to proceed for functional food development having antidiabetic attributes. Further in vivo studies may be carried out to support the findings of current study and to evaluate the toxicity.

### Funding
The authors received no funding for this work.

### Competing Interests
The authors declare that they have no competing interests.

### Author Contributions
- James William performed the experiments, contributed reagents/materials/analysis tools, approved the final draft, arrangement of equipment, calibration.

- Peter John analyzed the data, contributed reagents/materials/analysis tools, authored or reviewed drafts of the paper, approved the final draft.
- Muhammad Waseem Mumtaz conceived and designed the experiments.
- Ayoub Rashid Ch performed the experiments, contributed reagents/materials/analysis tools.
- Ahmad Adnan conceived and designed the experiments, authored or reviewed drafts of the paper, approved the final draft.
- Hamid Mukhtar analyzed the data, authored or reviewed drafts of the paper.
- Shahzad Sharif analyzed the data, prepared figures and/or tables.
- Syed Ali Raza performed the experiments, contributed reagents/materials/analysis tools, prepared figures and/or tables, statistical analysis, instrument handling, editting.
- Muhammad Tayyab Akhtar conceived and designed the experiments, analyzed the data.

## Data Availability

The raw data are available in the Supplemental Files.

## Supplemental Information

Supplemental information for this article can be found online at http://dx.doi.org/10.7717/peerj.7022#supplemental-information.

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
