# Peer review of "Antioxidant activity, α-glucosidase inhibition and phytochemical profiling of Hyophorbe lagenicaulis leaf extracts"

_PeerJ, doi:10.7717/peerj.7022_

## Round 0.1 · original submission · Major Revisions

Your manuscript must be revised to be Acceptable. When preparing your revised manuscript, you are asked to carefully consider the reviewer comments, which are attached, and submit a list of responses to the comments.

Reviewer 1 ·

Basic reporting

This manuscript by James Williams and his co-authors examines the hypothesis that”Leaf extract of Hyophorbe lagenicaulis exerts antioxidant and α-glucosidase inhibitory potential”. To this aim, they determined in vitro antioxidant and anti-diabetic activity of the extract using α-glucosidase assay. The article needs corrections that are mentioned in the following:

ABSTRACT
1. The abstract in not structured as background, methods, results and discussion rather it is presented in a single paragraph. The abstract should be structured as required by the journal.
2. No evidence is given in support of the statement that the identified compounds in the extract exert strong anti-oxidant and enzymatic effects in synergistic mode.
3. Spell mistakes are there in the abstract that should be reviewed carefully.

INTRODUCTION
1. The introduction would be strengthened if the authors were to more clearly describe how free radicals are generated in the body as a result of diabetes (mechanism of action). At the moment, the description is quite brief. As well, given that the authors are proposing DMT-2 might be reduced by controlling oxidative stress, there should be a fuller description of studies showing that this approach shows promise.
2. Three are so many grammatical mistakes in the introduction. Punctuations are not properly placed within the paragraphs. Should be revised carefully.

DISCUSSION
1. The authors need to more thoroughly discuss their results.
2. The presented results should be correlated with the management of DMT-2.
3. No statement of significance is mentioned in the manuscript.

REFERENCES
Reference style at the end is not according to the journal requirements.

Experimental design

1. No information is given about the collection of the plant material, identification and authentication. Voucher details are also missing.
2. Methodology is too brief to be understood by a common reader. Some more details are needed about the anti-oxidant activities protocols so that a common reader can get to know about the method.
3. No information is given that at what nm, absorbance was noted.
4. Α-amylase activity description is too brief. No information about the concentrations of the running solution.

Validity of the findings

RESULTS
1. The results are presented well. Figures’ legends must be in some detail.
2. Along with adopting a different approach to their analysis of statistical significance, the authors should also present a measure of practical significance; that is, they should report a standardized effect size (such as Cohen’s d value).

·

Basic reporting

- The manuscript was written in a clear and concise manner.
- Bibliographical references were provided in an appropriate manner.
- Figures need to be standardized. In figures 1, 2 and 3 the ethanolic extract should be written in the same way in all figures. Captions of figures are incomplete. The letters corresponding to the statistical analysis should be provided. For figure 4, information about peaks should be provided in the captions.
- The results are promising for the purpose of the study. However, few analysis about pharmacological properties were performed.

Experimental design

- The research is original. Is in accordance with the objectives of the journal.
- The research is well defined, relevant e meaningful, since diabetes type 2 is a disease that affects many individuals worldwide.
- The results are very preliminary. It seems that the antidiabetic potential of the extract is the focus of the work.There are other parameters that could support the antidiabetic capacity of this extract, such as in vitro glucose uptake assay, glucose diffusion inhibitory study, in vitro α-amylase activity and nonenzymatic glycosylation of haemoglobin. In this way, one or more of these tests could be performed to increment the results.
- The methodology was written very briefly. More details should be provided. For example, with respect to enzyme α-glucosidase evaluation, information about incubation time and buffer were not provided. In the same way, in the evaluation of the antioxidant parameters, few information was provided.

Validity of the findings

- The findings are interesting, however more analyzes should be carried out to better validate the conclusion obtained. The data are superficial to support the conclusions of the study.

Additional comments

Abstract:
- Cite the parameters evaluated in the antioxidant evaluation.
- In the conclusion is cited the following sentence "The Hyophorbe lagenicaulis might be an appropriate choice to manage diabetes mellitus and to develop formulations with antidiabetic properties". The results are superficials to lead to this conclusion.
- Add diabetes mellitus in key words.

Introduction:
- Write about the function of α-glucosidase. What carbohydrates are hydrolyzed for this enzyme?
- Remove this sentence "The UHPLC-Q-TOF-MS/MS was used to profile the metabolites in leaf extract of H. lagenicaulis.". This is metodology.

Results:
- In the method involving the reduction of molybdenum, the authors mention "This assay is widely used to evaluate the total antioxidant power of plant extracts and compounds (Prieto et al., 1999)." This is a very old reference. This method is currently used?
- Line 135: "Acarbose exhibited lowest IC-50 value (25.50 ± 0.45 μg/mL) being significantly higher than extracts.". IC-50 value of acarbose was lower than extract. The antioxidant activity was higher.

Discussion:
- The authors highlight the inhibitory activity of α-glucosidase by the extract. Some drugs are used in the treatment of DM and act by this mechanism, such as acarbose. These drugs commonly used in the treatment of DM present disadvantages and adverse effects? This should be mentioned since justifies the search for new drugs.
- What mechanism is associated with the antioxidant potential in the DPPH and molybdenum tests.

Conclusion:
- Experiments in vivo may be useful to suport this findings?

---

## Round 0.2 · Minor Revisions

As indicated by reviewer#1 the authors should redraw or re-upload figures 5 and 6 as they are currently unreadable.

Reviewer 1 ·

Basic reporting

no comments

Experimental design

no comments

Validity of the findings

The figure 5 & 6 are not visible , may be there is some problem in the online software program. The author is suggested to redraw or upload once again for clear visibility to the reader.

·

Basic reporting

Figures and legends have been corrected according to the suggestions.
The α-amylase inhibitor assay was added, which increased the pharmacological data.

Experimental design

The in vitro α-amylase activity has been added. More details about methodology has been added. The suggested changes were made.

Validity of the findings

The research is original. The research is well defined, relevant e meaningful, since diabetes type 2 is a disease that affects many individuals worldwide. The data provided by the manuscript are useful in this area of study

---

## Round 0.3 · accepted · Accept

The authors made teh requested corrections.

#